# Vaccination Campaign against Hepatitis B Virus in Italy: A History of Successful Achievements

**DOI:** 10.3390/vaccines11101531

**Published:** 2023-09-27

**Authors:** Tommaso Stroffolini, Giacomo Stroffolini

**Affiliations:** 1Department of Tropical and Infectious Diseases, Policlinico Umberto I, 00161 Rome, Italy; tommaso.stroffolini@hotmail.it; 2Department of Infectious-Tropical Diseases and Microbiology, IRCCS Sacro Cuore Don Calabria Hospital, Via Don A. Sempreboni, 5, Negrar, 37024 Verona, Italy

**Keywords:** HBV, vaccination, Italy

## Abstract

In Italy, the vaccination campaign against hepatitis B virus has been characterized by two phases. In the first phase (1984–1991), vaccination with plasma-derived vaccines was first recommended for the high-risk group. In the second phase (1991–nowadays), recombinant vaccine targeted, by law, infants 2 months old and teenagers 12 years old (limited to the first 12 years of campaign); screening for HBsAg became compulsory for all pregnant women during the third trimester of pregnancy. Successful achievements have been attained: No acute HBV case has been observed in the age group targeted by vaccination, the pool of chronic HBsAg carriers is strongly reduced, perinatal HBV transmission is under control, and acute delta virus hepatitis cases are nearly eliminated. The key point of this success has been the peculiar vaccination policy adopted. The combined vaccination of teenagers has generated an early immune cohort of youths, who are no longer at risk of acquiring HBV infection by sources of exposure (i.e., drug use and unsafe sex practices) typical of the young adults. Vaccination of household contacts with HBsAg-positive subjects represents an area of improvement; providing migrants and refugees access to healthcare services is also a focal point. In 2020, Italy became the first country in Europe to achieve the WHO’s regional hepatitis targets.

## 1. Introduction

Hepatitis B virus (HBV) is a significant global health problem. The World Health Organization (WHO) made an estimation in 2019 suggesting the presence of around 296 million individuals globally who were chronic carriers of the hepatitis B surface antigen (HBsAg). In this same report, WHO estimated 1.5 million newly recorded infections and unfortunate outcomes encompassing 820,000 deaths linked to cirrhosis and hepatocellular carcinoma (HCC) [1].

Following the identification of hepatitis B surface antigen (HBsAg) in 1965 [2], Blumberg spearheaded the development of the diagnostic test and the original vaccine. Initially termed the “Australia Antigen”, the virus received its name from a blood sample of an Australian aborigine that exhibited a reaction with an antibody present in the serum of an American hemophilia patient. In collaboration with Dr. Blumberg, microbiologist Irving Millman contributed to the development of a blood test targeting the hepatitis B virus. This test was integrated into blood banks’ screening processes in 1971, leading to a significant reduction in the risk of hepatitis B infections through blood transfusions [3,4]. Just four years after the discovery of the hepatitis B virus, Drs. Blumberg and Millman achieved another milestone by creating the inaugural hepatitis B vaccine. This initial vaccine variant involved a heat-treated version of the virus [4,5]. Subsequently, in 1981, the FDA granted approval for an enhanced plasma-derived hepatitis B vaccine designed for human use. Termed an “inactivated” vaccine, it involved the collection of blood from donors infected with the hepatitis B virus (HBsAg-positive). Through a series of meticulous steps, the pooled blood underwent processes to render the viral particles inactive, including treatments with formaldehyde and heat (referred to as “pasteurization”). Marketed as “Heptavax” by Merck Pharmaceuticals, this plasma-based vaccine stood as the pioneering commercial hepatitis B virus vaccine [3,4]. However, its usage was discontinued in 1990, and it is no longer accessible in the U.S.

In 1986, a new phase of research ushered in the advent of the second generation of genetically engineered (or DNA recombinant) hepatitis B vaccines. These innovative vaccines are synthetically produced and entirely devoid of blood components [4,5].

## 2. Background for Potential Vaccination Strategies

Enhanced by progressively more sensitive assays, the late 1970s witnessed the elucidation of mechanisms contributing to the pool of chronic HBsAg carriers. Notably, extensive evidence from the Far East revealed that infants born to HBsAg+/HBeAg+ mothers had an approximately 90% risk of acquiring infection at birth, with 85–95% of them progressing to chronic HBsAg carriage during their lifespan. In contrast, when mothers were HBsAg+/Anti-HBe+, the corresponding rates were notably lower, at around 25–30% and 20%, respectively [6,7], as shown in Table 1.

In regions at high endemic level, it was calculated that approximately 40% of carriers are established during the perinatal period [8,9]. Investigations conducted in Taiwan shortly thereafter demonstrated that the likelihood of chronicity after acquiring infection was 23% among preschool children [10] and diminished to 3% among university students [11]. This was one of the first pieces of evidence showing how different age groups, with their inherent different immunological status, would evolve differently after entering into contact with HBV. Subsequently, during the latter part of the 1980s, the introduction of serological testing for IgM anti-HBc (a reliable marker distinguishing recent from past HBV infection) provided conclusive evidence that acute clinical hepatitis B may lead to the chronic carrier status in fewer than 1% of immune competent adults [12]. As a matter of fact, the age at primary infection emerged as a pivotal factor influencing the risk of progressing into a chronic carrier state [13], as shown in Table 2.

These findings represented the groundwork for formulating potential future vaccine strategies applied to specific age groups and populations. Ultimately, in 1980, evidence was provided for a remarkable 93% efficacy of an inactivated hepatitis B (HB) vaccine in preventing HBV infection among homosexual men [14]. In particular, it was found that none of the vaccinees with a detectable immune response to the vaccine had clinical hepatitis B or asymptomatic antigenemia and that, within two months from the vaccination, 77% of the vaccinated persons had high levels of antibody against the hepatitis B surface antigen. It was also highlighted that this rate increased to 96% after the booster dose and remained essentially unchanged for the duration of the trial. These findings, coupled together with the emerging general knowledge on the topic, also provided one of the first pieces of evidence on the fact that the vaccine may be efficacious even when given after exposure [14].

One year later, pivotal evidence from Taiwan also firmly established the link between the HBsAg chronic carrier state and the occurrence of hepatocellular carcinoma (HCC) [15]. Notably, according to these data, HBV-infected individuals faced more than a 200-fold higher risk of developing HCC compared to uninfected controls.

Finally, in 1983, a significant breakthrough emerged, as immune prophylaxis employing hepatitis B immune globulins (IBIG) in combination with the hepatitis B vaccine demonstrated both safety and high efficacy in controlling the transmission of HBV infection from HBsAg-positive mothers to their newborns [16]. In particular, this trial found a combined efficacy of IBIG and vaccine of 94%. Also, persistent HBs antigenemia developed in only 6% of the 159 infants receiving prophylaxis but in 88% of the controls [16]. Inizio moduloFine modulo.

The comprehensive scenario became clearer, with pieces of evidence indicating that the HB vaccine could prevent acute HBV infection, chronic HBsAg carrier status, and even the newly discovered defective hepatitis Delta virus [17] (with the acquisition of this virus relying on the co-presence of HBsAg envelope for assembly and transmission [18]) along with hepatocellular carcinoma (HCC).

In this historical overview, we detail Italy’s evolving HBV vaccination policy and the resultant outcomes achieved.

## 3. First Phase (1984–1991)

In the late 1970s and early 1980s, Italy was a country at medium HBV endemic level, with a much higher rate of HBsAg prevalence in southern than in northern areas [19,20]; moreover, as many as 60% of chronic hepatitis cases were considered HBV-related [21].

As already briefly mentioned, the first vaccines, originating from France and the USA in the early 1980s, were plasma-derived [3,4,5]. These vaccines were in short supply and raised concerns of potential blood-borne virus contamination, including HIV. These concerns were later dismissed, as HIV susceptibility was deactivated during routine vaccine production [22].

Italy started vaccination in 1984, targeting high-risk groups like intravenous drug users, household contacts of chronic HBsAg+ carriers, newborns of HBsAg+-carrier mothers, homosexuals, healthcare workers, hemophiliacs, poly-transfused patients, and those on hemodialysis. Vaccination was offered free of charge and was just recommended at that time, without being compulsory for high-risk groups. In particular, the early decision to vaccinate health care workers was in agreement with contemporaneous U.S. data showing that HB vaccination decreased work-related infections with HBV [23].

While benefiting individuals, this targeted approach had limited effects on general population infection control and prevention. Plasma-derived vaccines were replaced in the late 1980s by recombinant DNA vaccines, offering extensive supplies and exceptional safety profiles [3,4,5].

## 4. Second Phase (1991–Nowadays)

The introduction of a recombinant DNA vaccine paved the way for a global expansion of vaccination efforts. The prospect of implementing a comprehensive vaccination campaign against HBV sparked debates within Italian scientific circles. Updated insights into HBV prevalence among a vast national sample of randomly selected individuals aged 3 to 19 were furnished by a study conducted in the late 1980s [24]. The study revealed an overall HBsAg and any HBV marker prevalence of 0.6% and 2.8%, respectively. These findings outlined a minimal virus exposure among the youngest generations.

The message coming from a second assessment of these data was clear: vaccination involving only the population of children 3 months old would have meant a waiting period of several years to observe an effective impact on the older generations. Adding to this policy a vaccination program involving adolescents would have provided a shorter time to obtain a great proportion of immune young adults.

In June 1991, a pivotal step was taken when vaccination became a legal requirement for offspring of HBsAg-carrier mothers. This policy followed mandatory third-trimester HBsAg screening for all expectant mothers irrespective of their country of birth. The vaccination directive extended to 2-month-old infants and children aged 12 years, in the latter category just for the first 12 years of the campaign. The age of 12 years was chosen because in Italy, attendance of primary school is compulsory by law, and children generally complete this level of schooling at around 11 or 12 years of age. Indeed, a high vaccine coverage could be expected selecting this age group, making it an ideal target for the campaign.

The vaccination program was fully effective in 1992. By 2004, vaccination for 12-year-olds was concluded, while it still continues for newborns of HBsAg-positive mothers and for 2-month-old infants. For the latter category, the vaccine schedule is three doses during the first year of life: at 2, 4, and 10 months of age. Indeed, newborns from HBsAg-negative mothers do not receive vaccine at birth, but they start the schedule at 2 months of age. Moreover, vaccination continues to be strongly recommended for high-risk groups (Figure 1).

The national healthcare system provided the vaccine at no cost for these individuals. This initiative has encompassed millions of people, achieving a remarkable coverage rate of 95% in children and adolescents [25]. Presently, nearly all individuals under 43 years of age are vaccinated against HBV in Italy. Immunizing teenagers proves valuable, albeit costly. It establishes defense against the virus prior to the start of high-risk behaviors, which are more common during adolescence, such as intravenous drug use and unsafe sexual intercourse. Remarkably, even in high-risk groups like healthcare professionals, vaccine coverage has notably risen from 64.5% in 1996 to 85.3% in 2006 [26], closely mirroring the decline in acute HBV cases within this cohort [27].

## 5. Evaluation of the Effectiveness of HB Vaccination Campaign

Recently, the effects of three decades of mandatory HB vaccination campaigns on acute HBV incidence and transmission patterns were evaluated [28]. Acute cases significantly dwindled in age groups targeted by the campaign, with zero instances in the 0–14 age group (100% reduction) and 0.1 cases per 100,000 inhabitants in the 15–24 age group (99.4% reduction). Furthermore, subjects older than 24 years of age experienced a substantial 87.5% reduction. The overall percentage of reduction was 92.0% (Figure 2).

Further evidence for the effectiveness of the HB vaccine was provided by the finding that acute HBV was observed in only 0.4% of vaccinated subjects during a 22-year (1993–2014) period of surveillance [29]. This latter finding suggests that in the real-world setting, the HB vaccine rarely is unable to generate immunity against HBV.

A significant impact has been observed concerning intravenous drug use’s role. While in the past, this was a major source of exposure for acquiring acute HBV, in the last decade, this risk factor is no longer linked to acute infection (OR 0.7: 95% C.I.: 0.5–1) [24]. Intravenous drug use contributed to 20.8% of acute cases in the 1991–1999 period, whereas it dropped to 3.3% during 2010–2019 [28]. These data differ from figures observed in USA, where a growing proportion of acute HBV cases among intravenous drug users has been reported in some states [30]. The relevance of intravenous drug use in the spread of HBV infection in USA was provided in another recent study: the prevalence of anti-HBc (the marker of previous exposure to the virus) has been stable since 2007 among people who denied drug use but almost doubled from 35.3% in 2001–2006 to 58.4% among those reporting this exposure [31]. This is particularly noteworthy in view of the current “national opioids crisis” that is affecting people in the USA.

The impact of HBV vaccine extends beyond the containment of acute infections; it has also considerably reduced the global burden of chronic liver diseases related to HBV. This statement is supported by several lines of evidence. Firstly, the proportion of pregnant women who tested HBsAg-positive during the third trimester consistently declined from 3% in 1985 [32] to 0.4% in 2009 [33] (Figure 3).

It is worth mentioning that the more recent survey (2009) found that no woman targeted by the vaccination policy tested positive for HBsAg [33]. Secondly, in a 2010 study conducted in a small town, no individuals below 30 years of age were anti-HBc-positive, reflecting no exposure to HBV [34].

Finally, the mean age of individuals with both chronic liver disease (CLD) and HBsAg positivity increased from 38.3 years in the 1980–1989 period [35] to 57.3 years in 2019 [36]. Notably, the prevalence of HBsAg among CLD cases also demonstrated a significant decline, plummeting from 61% to 12.1% in these two studies [35,36].

Italy’s management of perinatal HBV transmission is also showing positive results. As already pointed out, after 1984, HBsAg testing for pregnant women was recommended, later becoming compulsory by law in 1991. This change led to a substantial improvement in compliance with antenatal HBsAg testing, which progressively rose from 3.2% in 1984 [32] to 97.7% in 2009 [33] (Figure 4).

The perinatal transmission of HBV plays a pivotal role in sustaining the population of HBsAg carriers. HBsAg-positive women represent the reservoir of the virus among humans. They can transmit the infection to their newborns, replicating the cycle of transmission and assuring the persistence of the virus among humans in case of newborns of female sex. Thus, vaccination of these babies has the potential to disrupt the infection cycle even in high-income countries, where this mode of HBV transmission has lately contributed to a comparatively smaller pool of HBsAg carriers.

As already mentioned, in Italy’s most recent nationwide investigation on this matter [33], 97.7% of pregnant women underwent HBsAg testing. Moreover, 100% of infants from HBsAg-positive mothers were not missed for immunization, with no variation by country of birth. This latter aspect is of significant importance considering the increasing flow of immigration from regions with high HBsAg prevalence.

Overall, these findings highlight the minimal spread of HBV infection in Italy, implying that an effective virus control is nearing realization within a European country.

Countries that have implemented comprehensive HB vaccination campaigns have also reported exciting outcomes. Taiwan stands as an exemplary case of a previously highly endemic region, where substantial benefits resulted from the mass vaccination initiation in 1984. Notably, in Taipei, the prevalence of HBsAg in children plummeted from 9.4% in 1984 [37] to 0.4% in 2019 [38], and hepatocellular carcinoma (HCC) incidence among children aged 6–8 years dropped by 75% [39]. The universal vaccination of newborns successfully eradicated acute HBV and HCC among subjects under 20 years of age in Alaska: No cases of acute HBV infection and HCC have been reported in this age group since 1993 and 1995, respectively [40]. Finally, infant vaccination has reduced HBsAg prevalence among children in Gambia from 10% in 1986 to 0.6% in 1996 [41]. Taken together, these findings highlight the efficacy and reproducibility of the results of HB vaccination campaigns around the world in very different settings from high- to low- and middle-income countries.

## 6. Long-Lasting Vaccine Immunity

The success of universal HB vaccination is also related to the likelihood that vaccine-induced immunity may last for several years. Considering immunity, some questions may arise:-Firstly, how long may vaccination memory last?-Secondly, does the lack of a specific level of laboratory-confirmed protective anti-HBs titer over time still assure immunity?-Finally, is a booster dose required and, if so, in which moment?

From what has been evaluated, in immune-competent subjects, a reliable marker of protection is an anti-HBs titer value > 10 UI/mL measured 2–3 months after the last dose of the vaccination course [42]. Additionally, it has been established that the anti-HBs peak level after primary immunization correlates with the duration of protection, where the higher the peak, the longer the results [43].

The most compelling evidence for long-lasting immunity was recently provided in Alaska, where 86% of vaccinated people showed an anti-HBs-protective level after priming [44]. Similar results were previously shown in Taiwan after a period of 25 years [45] and in the Gambia [46] as per evaluation up to 24 years after the vaccination campaign.

It has also been established that the antibody titer level progressively declines over time; however, its loss does not imply a full risk for the loss of protection. The immune memory persists in most vaccinated subjects, who are able to generate an efficient anamnestic response when naturally exposed to HBV or exposed to a booster [47,48,49,50]. Indeed, in immune-competent individuals, an effective long-term protective effect may persist despite the loss of humoral immunity, as measured by antibody titer in blood.

Currently, the European Association Study of Liver (EASL) does not recommend a booster dose in the absence of other clear data that would suggest doing so [51].

## 7. The Role of Antiviral Therapy

Despite the substantial reduction in the total number of HBsAg-positive subjects in Italy, a notable fraction of carriers still persists. Most of these individuals are now above the age of 50 years and have harbored the HBV-carrier state for a long period, having contracted the infection during their younger years. While vaccines can prevent infection in susceptible individuals, they are not effective for those already infected.

For such cases, third-generation nucleos(t)ide analogues, specifically entecavir and tenofovir, introduced since the first half of the 2000s, offer a valuable and efficacious solution. Both drugs can suppress viral replication, halting disease progression and curbing infection transmission from carriers to susceptible individuals [51]. Regrettably, these drugs are unable to expunge integrated HBV DNA sequences from infected hepatocytes [51]. Recent studies have indicated that a small proportion of the cccDNA reservoir is continually replenished through sustained low-level HBV replication, while a substantial portion of cccDNA endures over time [52].

The potential emergence of novel drugs capable of eliminating the genomic HBV reservoir (including cccDNA and integrated HBV DNA) [53]) may constitute another stride toward potentially halting disease progression.

## 8. The Impact of HB Vaccination on HDV Infection

To date, a vaccine designed expressly against the HDV virus is absent. As previously alluded to, HDV only manifests in the context of HBsAg co-presence due to its dependency on HBV for replication [18]. The HB vaccine, which curbs the population of chronic HBsAg carriers, was anticipated to concurrently diminish HDV endemicity by depriving the virus of its required biological platform for dissemination.

This anticipation has materialized in reality, as outlined by the decline in acute Delta hepatitis incidence in Italy from 3.2 cases per 1 million people in 1987 to 0.04 cases in 2019 [54], paralleling the trajectory of acute HBV per 100,000, which was lowered from 10.0 to 0.39 cases during the same period [28].

## 9. Areas of Improvement and New Challenges

Given the present context, it is essential to explore not only potential areas for enhancement but also the novel challenges that are coming to the forefront.

Firstly, attention should be specifically addressed to the vaccination of household contacts of subjects that are known to be HBsAg-positive. Despite being a well-recognized high-risk group, and although strong recommendations for this high-risk population have been in effect since 1984, this group often misses the HB vaccine. As per available data, in Italy, this mode of exposure currently represents the most efficient manner for acquiring HBV infection (OR 10.8; 95% C.I. = 7.8–14.9) [28]. Interestingly, vaccination was missed in as many as 40% of acute HBV cases belonging this high-risk group despite awareness of their cohabitants’ carrier status. Many of these household contacts are unaware of their heightened risk of HBV infection and consequently underestimate the associated danger. To address this high-risk group more proactively, general practitioners (GPs) and specialists that take care of HBsAg+ patients should provide more comprehensive counseling.

Additionally, achieving a testing rate of 97.7% for pregnant women screened for HBsAg during their third trimester is commendable from a public health perspective. However, the paramount goal of preventing perinatal HBV transmission motivates us to strive for a 100% testing coverage rate among pregnant women.

Finally, HBV infection in migrants poses a fresh challenge for Italy and other European and high-income countries. These individuals mostly come from areas where HB vaccination coverage is suboptimal. As they are often not immunized against HBV, they may acquire the infection through risk behaviors and, once infected, spread the virus to susceptible hosts. As per available data, it has been calculated that migrants account for one-fifth of all acute HBV cases occurring in Italy and that they are three-fold more likely (95% C.I.: 2.5–3.6) to acquire acute infection than Italian natives [25]. An advisable preventive strategy could be one involving a more proactive testing and offering HBV vaccines to susceptible migrants [55]. In particular, recommendations agree on the fact that HBV screening should be offered to all migrants and asylum seekers coming from high or medium endemic countries (HBsAg prevalence ≥ 2%), for those with specific risk factors, and for all pregnant women [55].

Furthermore, migrants predominantly originate from areas at a high HBV endemic level. The percentage of individuals from non-Italian countries having a chronic HBV infection accounted, in 2019, for a more than four-fold increase compared to 2001 (19.0% vs. 4.5%) [36]. For this segment of population, recommending antiviral treatment with comprehensive counseling to curb further infection spread is highly advisable. In 2008, a comprehensive study concerning 3305 chronic carriers of HBsAg showed that the condition of the migrant was associated with a lower likelihood of access to antiviral treatment [56]. However, with an 11-year gap [40], nowadays, the status of being a migrant is no longer a barrier (O.R. 2.1; 95% CI: 0.9–4.4) for antiviral therapy. In Italy, public hospitals and effective HBV drugs are freely accessible to all.

## 10. Conclusions and Perspectives

Over the course of four decades, Italy’s HB vaccination history has been marked by significant accomplishments. Acute hepatitis B cases are no longer reported among those targeted by vaccination. The population of chronic HBsAg carriers has drastically decreased, and the transmission of perinatally acquired HBV is effectively managed. The occurrence of acute HDV cases is nearly eradicated. This remarkable success can be attributed to the distinctive vaccination policy adopted, which added compulsory vaccination for adolescents to the childhood vaccination program. This strategy has protected cohorts of young adults against the risk of acquiring HBV infection by sources of exposure typical of this age group.

Despite its the expense it incurs, this program has proven cost-effective. This favorable trajectory is anticipated to continue as universal vaccination expands the immune cohorts annually, and the existing pool of HBsAg carrier ages dwindles in numbers over time. However, a notable area for improvement remains the HB vaccination coverage for household contacts of chronic HBsAg carriers. Another significant challenge stems from the influx of migrants from HBV endemic regions.

Echoing the World Health Organization’s (WHO) recommendation [57], several countries have enclosed the HB vaccine in the schedule of infant vaccinations. Italian achievements are expected to be replicated in other countries in the future, contingent upon their capacity to support disadvantaged populations. Limited access to healthcare, particularly affecting migrants and asylum seekers due to socioeconomic factors, is a challenge seen in certain high-income countries, in particular North America [58]. In this context, migrants face barriers to healthcare due to low income and lack of insurance coverage [59]. In Europe, including countries adjacent to Italy, such as France and those in the Balkans, significant migration patterns have been observed in recent decades. In France, compulsory adolescent vaccination was introduced in addition to infant vaccination in 1994 [60]. While it initially showed promise, adolescent vaccine coverage decreased to 55.4% by 2011 [61]. In the Balkan region, where mandatory infant vaccination is only in place in Bulgaria and Croatia, information regarding the prevalence of chronic hepatitis B infection remains incomplete due to limited screening and surveillance efforts [62].

As of January 2020, Italy received a commendation from the WHO ETAGE (European Technical Advisory Group of Experts) for these successful outcomes [63].

Italy’s forthcoming priorities encompass maintaining HBsAg testing for pregnant women in their third trimester, with subsequent immunization of newborns of positive mothers. Universal infant vaccination and providing migrants and refugees free access to healthcare services are also focal points. Ultimately, the preservation of the current welfare state is of paramount significance.

## Figures and Tables

**Figure 1 vaccines-11-01531-f001:**
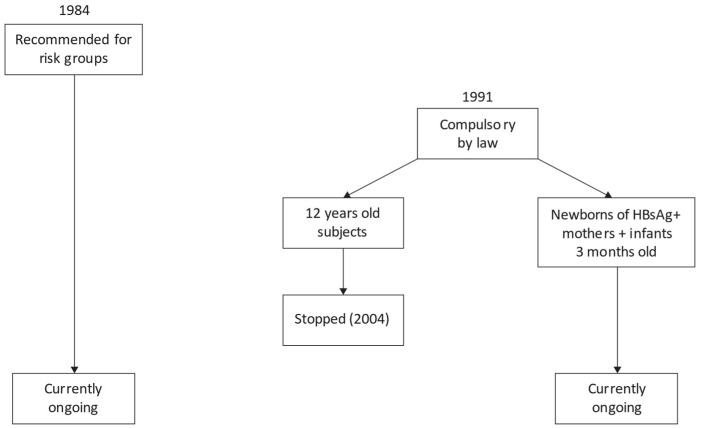
Flow-chart of HB vaccination policy in Italy.

**Figure 2 vaccines-11-01531-f002:**
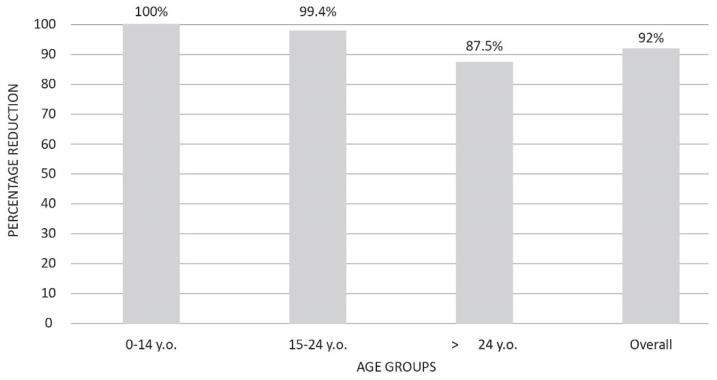
Percentage reduction of the incidence of acute HBV cases in Italy in the year 2019 compared to the year 1990 by age groups (adapted from reference [28]).

**Figure 3 vaccines-11-01531-f003:**
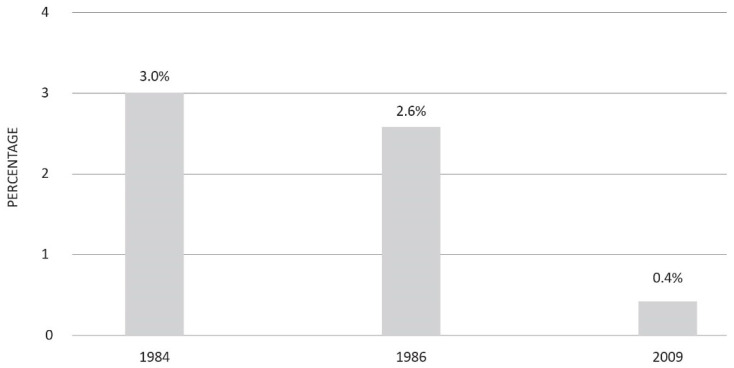
HBsAg prevalence among Italian pregnant women before and after the introduction of compulsory HB vaccination (adapted from references [32,33]).

**Figure 4 vaccines-11-01531-f004:**
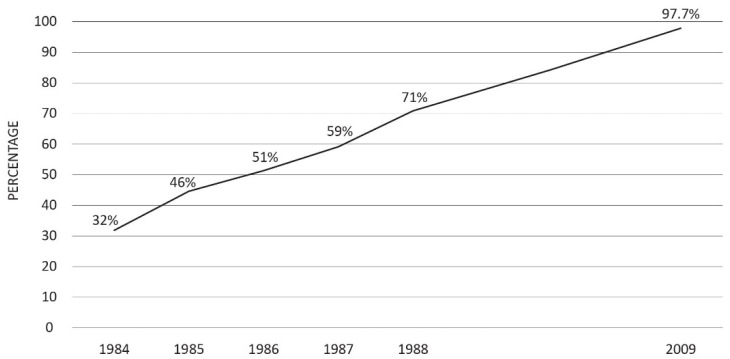
The proportion of pregnant women tested for HBsAg during the third trimester of pregnancy in Italy is shown as increasing over time (adapted from references [32,33]).

**Table 1 vaccines-11-01531-t001:** Vertical transmission of HBV by serological status of mother (adapted from references [4,5]).

Mother HBV Serology	Newborn	
	Attack Rate	Risk of Chronicity
HBsAg+/HBeAg+	90.0%	90.0%
HBsAg+/Anti-HBe+	25.0%	20.0%

**Table 2 vaccines-11-01531-t002:** Age at infection is a crucial point for the risk of becoming a chronic HBsAg carrier.

90%	Newborns of HBsAg+/HBeAg+ Mother	(Refs. [2,3])
23%	Preschool children	(Ref. [6])
3%	University students	(Ref. [7])
−1%	Immune competent adults	(Ref. [8])

## Data Availability

Not applicable.

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
