# Peer review of "Vaccination Campaign against Hepatitis B Virus in Italy: A History of Successful Achievements"

_vaccines, 2023, doi:10.3390/vaccines11101531_

Round 1

Reviewer 1 Report

Dear Authors

It is a nice summarey on HBV vaccination in Italy

It could further be imrpoved if you could include the following suggestions

1. Try to present the intervention and its effect in pictorial manner

2. INtroduction section is very long, it can easily be  reduced 

3. What was the effect of vaccination on HBV seroprevalence in high-risk population such as people living with HIV, people on dialysis etc

4. Try to add some data on cost-effectiveness and the number of lives saved with the intervention

Language need imrpovement

Author Response

Dear Reviewer

I would like to express my sincere gratitude for taking the time to review our manuscript titled "Vaccination campaign against Hepatitis B virus in Italy: a history of successful achievements" submitted to Vaccines. Your thoughtful and insightful comments have been instrumental in improving the quality and rigor of our research. We highly value your expertise and the constructive feedback you have provided.

In response to your comments, we have carefully addressed each of them as follows:

1) It has been previously reported in Figure 1. 2) Introduction section has been reduced by some paragraphs and splitted in two parts. 3) Regarding the current research, there is a dearth of studies conducted on this particular subject.

4) There is a scarcity of such estimates. This can be attributed to their inherent complexity, as they are exceedingly challenging to calculate due to numerous confounding factors that can obscure the analysis.

We believe that the revisions made in response to your feedback have significantly enhanced the quality and validity of our research. Once again, we extend our appreciation for your valuable input

Reviewer 2 Report

The manuscript presented to be published as a review represents a very interesting and informative story about the success of vaccination campaign against Hepatitis B virus in Italy, which has been characterized by two phases. In addition, to the introduction part, where the authors present an overview of the Hepatitis B and its worldwide distribution, they also describe in the other parapgraphs the successful achievements, which have been attained in Italy, like no acute HBV case was observed in the age group targeted by vaccination, the strongly reduced pool of chronic HBsAg carriers and acute delta virus hepatitis cases, that resulted nearly eliminated. The other sections are well documented by including the graphics and generally it is appropriately written. 

Anyway, I would recommend including in the broad analyses, the cases of the neighboring countries, like, France, or even South-Western Balkan countries, from which they have had a lot of immigrants during the last 30-40 years.

Author Response

Dear Reviewer, I would like to express my sincere gratitude for taking the time to review our manuscript titled "Vaccination campaign against Hepatitis B virus in Italy: a history of successful achievements" submitted to Vaccines. Your thoughtful and insightful comments have been instrumental in improving the quality and rigor of our research. We highly value your expertise and the constructive feedback you have provided. The cases of the neighboring countries have been broadly analysed. Once again, we extend our appreciation for your valuable input

Reviewer 3 Report

Abstract,

Add the way forward

Introduction

Line 25—27

Comment: Provide reference the below facts

Hepatitis B is a viral infection that primarily affects the liver. It is caused by the hepatitis B virus (HBV), which belongs to the Hepadnaviridae family. Hepatitis B can range in severity from a mild illness lasting a few weeks to a serious, lifelong condition. Hepatitis B (HB) is a significant global health problem.

Line 28

In fact, the World Health Organization (WHO) made an estimation in 2019

Comment: Provide latest figure up to 2022

Line 225

As already mentioned, in Italy's most recent nationwide investigation on this matter [32], 97.7% of pregnant women underwent HBsAg testing.

Comment: There is still 2% unscreened how would you go about investigate them?

The authors could consider this is an areas of improvement and new challenges

General comment.

Could comment and add if birth Hepatitis B vaccine is given and the impact if any?

Provide information on the current hepatitis B schedule

The manuscript has been written in good English

Author Response

Dear Reviewer, I would like to express my sincere gratitude for taking the time to review our manuscript titled "Vaccination campaign against Hepatitis B virus in Italy: a history of successful achievements" submitted to Vaccines. Your thoughtful and insightful comments have been instrumental in improving the quality and rigor of our research. We highly value your expertise and the constructive feedback you have provided. - The way forward has been added in the abstract. - Lines 25-27. According also to other reviewers' commentaries, paragraph has been eliminated. - Line 28. Latest figures and estimates from WHO are available until to 2019. - Line 225. A rate of only 2.3% still unscreened pregnant women is an excellent result from the public health point of view. Anyway, we made additional considerations in "areas of improvement". - Birth hepatitis B vaccine is given only to newborns of HBsAg positive mother, togheter with IBIG. - The current hepatitis B schedule is thre doses during the first year of life: 2, 3, and 10 months of age. Once again, we extend our appreciation for your valuable input

Reviewer 4 Report

This paper reads like an interesting chapter in the history of the discoveries of HBV and vaccines to protect against the infection. The authors did not  mention the early decision to vaccinate healthcare workers, which resulted in a decrease of work-related infection with HBV (Osterholm MT, Garayalde SM. Clinical viral hepatitis B among Minnesota hospital personnel. Results of a ten-year statewide survey. JAMA. 1985 Dec 13;254(22):3207-12.). Another point is the reference #27, which is actually a second description of the data in this paper.

This paper reads like an interesting chapter in the history of the discoveries of HBV and vaccines to protect against the infection. The authors did not  mention the early decision to vaccinate healthcare workers, which resulted in a decrease of work-related infection with HBV (Osterholm MT, Garayalde SM. Clinical viral hepatitis B among Minnesota hospital personnel. Results of a ten-year statewide survey. JAMA. 1985 Dec 13;254(22):3207-12.). Another point is the reference #27, which is actually a second description of the data in this paper.

Author Response

Dear Reviewer, I would like to express my sincere gratitude for taking the time to review our manuscript titled "Vaccination campaign against Hepatitis B virus in Italy: a history of successful achievements" submitted to Vaccines. Your thoughtful and insightful comments have been instrumental in improving the quality and rigor of our research. We highly value your expertise and the constructive feedback you have provided. The early decision to vaccinate healthcare workers has been mentioned, and reference included. Once again, we extend our appreciation for your valuable input 

Round 2

Reviewer 1 Report

THanks for your response

It is ok